# Impact of State Stroke Systems of Care Laws on Stroke Outcomes

**DOI:** 10.3390/healthcare11212842

**Published:** 2023-10-28

**Authors:** Erika B. Fulmer, Dana Keener Mast, Lucas Godoy Garraza, Siobhan Gilchrist, Aysha Rasool, Ye Xu, Amanda Brown, Nina Omeaku, Zhiqiu Ye, Bruce Donald, Sharada Shantharam, Sallyann Coleman King, Adebola Popoola, Kristen Cincotta

**Affiliations:** 1Division for Heart Disease and Stroke Prevention, National Center for Chronic Disease Prevention and Health Promotion, Centers for Disease Control and Prevention, 4770 Buford Highway, NE, Mailstop MS-S107-1, Atlanta, GA 30341, USA; aysharasool@outlook.com (A.R.); sophieye999@gmail.com (Z.Y.); ktq4@cdc.gov (S.S.); fjq9@cdc.gov (S.C.K.); nci9@cdc.gov (A.P.); 2ICF, 1902 Reston Metro Plaza, Reston, VA 20190, USA; dana.keenermast@icf.com (D.K.M.); lucas.godoygarraza@icf.com (L.G.G.); ye.xu@icf.com (Y.X.); kristen.cincotta@icf.com (K.C.); 3ASRT, Inc., 4158 Onslow Place SE, Smyrna, GA 30080, USA; sgilchrist@mitre.org (S.G.); vsw7@cdc.gov (A.B.); yom4@cdc.gov (N.O.); qsq3@cdc.gov (B.D.); 4Oak Ridge Institute for Science and Education, P.O. Box 117, Oak Ridge, TN 37831-0117, USA

**Keywords:** stroke systems of care, law, health policy, public health, legal epidemiology, stroke out comes

## Abstract

Since 2003, 38 US states and Washington, DC have adopted legislation and/or regulations to strengthen stroke systems of care (SSOCs). This study estimated the impact of SSOC laws on stroke outcomes. We used a coded legal dataset of 50 states and DC SSOC laws (years 2003–2018), national stroke accreditation information (years 1997–2018), data from the Healthcare Cost and Utilization Project (years 2012–2018), and National Vital Statistics System (years 1979–2019). We applied a natural experimental design paired with longitudinal modeling to estimate the impact of having one or more SSOC policies in effect on outcomes. On average, states with one or more SSOC policies in effect achieved better access to primary stroke centers (PSCs) than expected without SSOC policies (ranging from 2.7 to 8.0 percentage points (PP) higher), lower inpatient hospital costs (USD 610–1724 less per hospital stay), lower age-adjusted stroke mortality (1.0–1.6 fewer annual deaths per 100,000), a higher proportion of stroke patients with brain imaging results within 45 min of emergency department arrival (3.6–5.0 PP higher), and, in some states, lower in-hospital stroke mortality (5 fewer deaths per 1000). Findings were mixed for some outcomes and there was limited evidence of model fit for others. No effect was observed in racial and/or rural disparities in stroke mortality.

## 1. Introduction

In 2020, stroke was identified as the fifth leading cause of death in the United States (US), accounting for over 160,000 deaths [1]. Stroke is a leading cause of long-term disability in the US and results in nearly USD 57 billion in medical expenses, including direct medical costs and indirect costs from premature mortality [2]. Lifesaving, disability-reducing treatments are available for stroke. However, to be effective, patients must receive care within the first hours of stroke onset [3].

Public health practitioners can create a supportive infrastructure for ensuring that stroke patients have time-sensitive access to lifesaving treatments. Comprehensive stroke systems of care (SSOCs) are integrated networks (often coordinated at state and/or regional levels) helping stroke patients receive the most appropriate and timely treatment [3]. SSOCs are designed to support and enhance all phases of stroke care including prehospital identification, ambulance transport, stroke center certification, in-hospital diagnosis and treatment, and coordinated transitions from hospital to home, supporting secondary prevention, rehabilitation, and recovery [3].

States have applied legislation to create and improve SSOCs. As of 2018, 38 states and DC enacted state SSOC laws to coordinate and organize local health services to improve stroke outcomes [4]. These laws have changed over time. For example, in January 2012, there were eight states with legislation in place to triage and transport suspected stroke patients to certified primary stroke centers (PSCs). In January 2015, there were nine states with the same legislation, and by January 2018, the number had grown to sixteen states (see CDC’s National Environmental Public Health Tracking Network for additional information, https://ephtracking.cdc.gov/DataExplorer/#/ (accessed on 10 August 2023)).

Early evidence suggests that SSOC policy interventions can improve the quality and efficiency of care [5,6,7]. National organizations, including the American Heart Association/American Stroke Association (AHA/ASA) and the Brain Attack Coalition (BAC), have recommended SSOC policies to promote more efficient and effective stroke treatment [3,8,9,10]. Additionally, during the past 20 years, recommendations and funding provided by AHA/ASA, BAC, and CDC’s Coverdell Program have prompted the use of measures that pertain to both quality and efficiency of stroke care and health outcomes [3,9,10]. AHA/ASA Get With The Guidelines (GWTG)-participating hospitals and Coverdell Program recipients have demonstrated improvements in the timeliness and administration of thrombolytic treatment, and for GWTG, improved clinical outcomes [11,12]. Many states have used their legal authorities to develop and implement SSOCs through task forces, state regulatory agencies, and regional entities [4].

As of 2023, no previous studies have measured the impact of state-level SSOC laws on stroke outcomes using a national dataset. This study contributes to the evidence base for public health policy by addressing policy outcomes [13]. Using a natural experimental design [14], we modeled longitudinal data from secondary sources to estimate the impact of state SSOC laws in effect between 2002 and 2018 on outcomes related to quality and efficiency of stroke care and stroke health outcomes across 50 states and DC.

## 2. Materials and Methods

A CDC human subjects review determined that this study did not constitute human subjects research and, therefore, did not require full review. The data that support the findings are available from the corresponding author upon reasonable request.

### 2.1. State SSOC Law Classification

Two legal analysts created a longitudinal legal dataset of SSOC laws. They independently searched WestlawNext for relevant state policies (including statutes, regulations, session laws, and uncodified legislation) in all 50 states and DC for each year between 2002 (2003 is the first year that SSOC policies are addressed in state law) and 2018 (the SSOC legal dataset is available on CDC’s National Environmental Public Health Tracking Network). Data on relevant laws were collected at the state level only and did not account for any stroke policies passed at the county, city, or township levels. Researchers independently coded laws in effect (per state and year) for the presence of 19 SSOC policy intervention variables as either “required”, “required with exceptions”, “authorized”, or “silent”. The researchers then compared the independently coded records and resolved differences. For states that expressly required an executive agency to promulgate regulations to implement enabling legislation, the effective date of the regulations was used to determine the first year the policy intervention was in effect.

Using the coded data, a dummy variable was created reflecting each state’s SSOC policy intervention status in a specific year. When a state had at least one SSOC policy intervention in effect (either required, required with exceptions, or authorized) on 1 January, the state was coded as “1” for that year; when no SSOC policy interventions were in effect on 1 January, the state was coded as “0” for that year. Because of the large number of policy interventions, the complexity of the methods, and the fact that some states have no stroke laws in effect, the current analysis focuses on whether states with one or more policies in effect demonstrate better outcomes over time than they would have without any state SSOC policies. Future analyses are needed to compare the relative contribution of specific SSOC policies. In total, 38 states and DC had at least one SSOC policy intervention in effect during the study period. Policy interventions, descriptions, and states with policy in effect are listed in Table 1.

### 2.2. Outcome Metrics and Data Sources

Measures related to quality and efficiency of stroke care as well as stroke health outcomes were examined at the state level using all years of historical data available to maximize model fit. The number of states and years of data varied for each outcome based on the availability of data. Selection of measures was informed by a conceptual model demonstrating the theory of change for state SSOC [15].

Certified Primary Stroke Centers (PSC). PSCs are hospital-based centers with re- sources, capabilities, and protocols in place to care for acute stroke patients [16] and are certified by three ASA recognized national accreditation organizations, including The Joint Commission (TJC), Healthcare Facilities Accreditation Program (HFAP), and Det Norske Veritas (DNV). Annual certification data were curated from these organizations, as well as from the Centers for Disease Control and Prevention (CDC), Division for Heart Disease and Stroke Prevention and the Cleveland Clinic, Cerebrovascular Center, Neurological Institute, and Department of Qualitative Health Sciences [7] to determine the total number of certified PSCs in each state and DC for the years 2007–2018; data were not available for 2008. To ensure uniformity in the stroke center standards, we did not include stroke centers accredited at the state level. For each state and year, the number of PSCs was divided by the total number of hospitals in the state to calculate the proportion of certified PSCs per state and year.

Brain scan imaging and interpretation within 45 min of emergency department arrival. The percentage of patients experiencing stroke symptoms who received brain scan results within 45 min of arrival at an emergency department by state and year was obtained from Hospital Compare. Data were available for all 50 states and DC for the years 2012 to 2018. DC was removed from the analysis as an outlier due to extreme values.

In-hospital stroke costs. Aggregated in-hospital charges for stroke patients were available for 37 states from the Healthcare Cost and Utilization Project (HCUP) for the years 2006–2017. Charges were converted to costs using the cost-to-charge ratio (in dollars) for stroke patients by year and state. Of states with available in-hospital stroke cost-related data, 27 states had at least one SSOC policy intervention in effect.

In-hospital stroke mortality. In-hospital stroke mortality is defined as the percentage of stroke inpatients who died while hospitalized. State-level aggregates for this metric were available from HCUP for 37 states from 1997–2017. Of states with available in-hospital stroke mortality data, 27 states had at least one SSOC policy intervention in effect.

Age-adjusted stroke mortality. Annual age-adjusted death rates due to all types of stroke (International Classification of Diseases, Ninth Revision (ICD-9) codes 430–434 and 436–438 for years 1979–1998; and ICD-10 codes 160–169 for years 1999–2019) per 100,000 population were obtained for all 50 states and DC from the National Vital Statistics System (NVSS) for the years 1979–2019.

Racial disparities in stroke mortality. Using data obtained from NVSS, we computed the absolute difference in age-adjusted stroke mortality rates between White (defined as non- Hispanic White) and non-White (defined as Hispanic and non-Hispanic non-White) populations by state and year. NVSS data were available for 46 states from 1999–2019. Of the states with available racial disparities in stroke mortality data, 38 states had at least one SSOC policy in effect.

Rural/urban disparities in stroke mortality. Using data obtained from NVSS, we calculated the absolute difference in age-adjusted stroke mortality rates between rural (de fined as nonmetropolitan areas within the 2013 National Center for Health Statistics’ Urban–Rural Classification Scheme for Counties) and non-rural (defined as all metropolitan levels) counties by state and year. NVSS data for rural and non-rural counties were available for 47 states for the years 1999–2019. Of the states with available rural/urban disparities in stroke mortality data, 35 states had at least one SSOC policy in effect.

### 2.3. Measured Covariates

To control for factors that may be associated with stroke outcomes other than state SSOC laws, we obtained state-level metrics to serve as covariates in the analysis. Measured covariates included stroke incidence rate measured by the Behavioral Risk Factor Surveillance System [17]; percentages of population classified as rural, non-White, and over 60 years of age from the U.S. Census Bureau [18]; Medicaid expansion status [19]; funding status from CDC’s Paul Coverdell National Acute Stroke Program [20]; reduced number of hospital beds due to hospital closures [21]; and geographic census region [22]. In addition, mean age of stroke patients was obtained from HCUP as a covariate for in-hospital stroke cost and in-hospital stroke mortality [23]. We included “door-to-diagnostic evaluation by qualified medical personnel” from Hospital Compare as a covariate for brain imaging rates within 45 min to control for secular improvements in hospital care [24]. We also included the frequency of hospital admissions (a state-level metric obtained from HCUP) as a covariate for proportion of certified PSCs, in-hospital stroke cost, and in-hospital stroke mortality.

## 3. Statistical Analysis

We used R (Version 4.2.1) to develop Bayesian Additive Regression Trees (BARTs) to estimate the impact of having at least one SSOC policy intervention in effect on each outcome examined [25,26,27,28]. New policies take time before they are fully implemented and achieve desired effects [29]. To account for the lag between the time a policy goes into effect and its potential impact, we estimated the effect starting a minimum of one year after the first SSOC policy intervention(s) were in effect in each state. To determine this, we examined whether the policy was in effect on 1 January of that year; however, effective dates can occur throughout the year, so some laws may have been in effect multiple months before applying the one-year lag. We measured the counterfactual by setting each outcome to “missing” in each state and year when at least one SSOC policy was in effect (except for the first year).

A BART model was fitted to the remaining outcome data in each state for all years when no policies were in effect until one year after the first policy went into effect (including all years of data for 12 states where SSOC laws were never in effect). The model included all the measured covariates, flexibly incorporating nonlinear relationships and/or interactions when present. To induce regularization (a tool used to avoid overfitting the model and increase out-of-sample predictive performance), we used the default set of prior distribution suggested by Chipman and colleagues [25].

The fitted model was used to predict the outcome values in the years set to “missing” (the counterfactual outcome) [30]. Finally, we measured the difference between the predicted and observed outcomes in states with SSOC laws to produce annual state-specific estimates of the effect for each year an SSOC policy intervention was in effect (excluding the first year).

Controlling for potential confounders. In addition to the measured covariates already mentioned, the BART models also used patterns in the historical outcome data to account for unmeasured covariates. Specifically, state and year identifiers (i.e., a set of indicator or dummy variables, one for each state and one for each year) were included as predictors in the model to capture state and year “fixed effects”, including differences between states that are relatively constant over time (e.g., hospital infrastructure) and temporal changes that affect all states in a similar way (e.g., changes in federal regulation). The model allowed for interactions between the state and year fixed effects to reflect state-specific trends [31]. Finally, we included an estimate of the probability that at least one SSOC policy was in effect for each state and year as a function of the same covariates listed above. While the estimated propensity is a transformation of covariates already included in the model, including an estimate of the propensity score as a covariate is recommended to guard against confounding that can be induced by regularization, particularly in the context of a large number of predictors relative to the number of observations [25].

After considering the available historical information on each outcome, as well as measured and unmeasured covariates to the greatest extent possible, we expected no systematic differences between states other than the SSOC policy interventions that are in effect. To the extent that we were able to account for state-specific trends and programmatic contextual factors, we infer that the difference between the observed and the predicted outcomes measure the effect of having any SSOC policy intervention in place.

Estimating the aggregate-level effect. Using the state-specific estimates of the effect on each outcome, we averaged the state-year specific estimates across groups of states and time to compute the aggregated summary for each outcome. Given that states’ first SSOC policy interventions became effective in different years, the number of years that the estimated effect is available after the first policy change varies by state (i.e., states that enacted SSOC policies in earlier years have more available years of the estimated effect). As such, the number of states with data that can be used to summarize the effect of having at least one SSOC policy decreases as the number of years since the first policy was in effect increases.

To balance the number of years and states examined in each aggregated summary and to minimize changes in the number of states included in the estimate, we computed the aggregated effect at three segments (i.e., quartiles) of the distribution of states by the number of years the estimated effect was available per state. Each increasing quartile included a smaller number of states and a higher number of years contributing to the estimated aggregated effect. At the first quartile (25%) of the distribution, approximately three-quarters of the states with SSOC policies in effect contributed information to the average, including a mix of “early” and “late” adopters. At the third quartile (75%) of the distribution, approximately one-quarter of the states contributed information to the aver- age representing only the “early” adopters with a longer span of available data. Importantly, within each quartile examined for a particular outcome, the same sample of states was used to determine the contribution of information to the aggregated summary.

Assessing the robustness of the model. We used an equivalence test to measure how well the predictive model reproduces the outcome before any SSOC policy interventions were in effect (or in the case of states without any policies, during the entire period) [32]. The average discrepancy between the estimated and actual outcome before any policy was in effect should be close to zero. The notation “Pr(|res| > 2.5%” indicates the posterior probability (i.e., the distribution of predicted values after accounting for observed data) of observing a discrepancy that is larger than +/−2.5% between the predicted and observed values; small values indicate strong historical fit.

## 4. Results

### 4.1. Certified Primary Stroke Centers

Twenty-two states had between four and nine years of PSC certification data starting one year after SSOC policies were in effect. Among twenty-two states in the first quartile with at least four years of data, the average proportion of certified PSC hospitals was 2.7 percentage points (PP) higher than predicted in the absence of any SSOC policies (Figure 1A). The second quartile of 12 states with at least seven years of PSC data had an average proportion of certified PSCs that was 5.1 PP higher than predicted (Figure 1B). In the third quartile of seven states with at least nine years of PSC data after SSOC policies were in effect, the average proportion of certified PSCs was 8.0 PP higher than predicted in the absence of SSOC policies (Figure 1C). The 90% credible interval (CI) (i.e., the central portion of the posterior distribution that contains 90% of the values) of the average discrepancy for each group of states at four, seven, and nine years after the policy went into effect did not cross zero, lending strong evidence for the estimated effect. The posterior probability of a large discrepancy between observed and predicted values in years before any SSOC policies were in effect was very low for all three quartiles of states. Average differences, CIs, posterior probability residuals, and placebo checks for all outcome metrics are provided in Appendix A.

### 4.2. Brain Imaging Results within 45 Min

Fourteen states had between three and five years of brain imaging data starting one year after their first policy intervention took effect. Among the first quartile of states with at least three years of brain imaging data, the average observed proportion of stroke patients that received brain imaging results within 45 min of emergency department arrival was 3.6 PP higher than the predicted outcome in the absence of any SSOC policies (Figure 2A). For the second quartile of 11 states with at least four years of outcome data, the brain imaging rate was 3.6 PP higher than the predicted outcome (Figure 2B). For the six states in the third quartile with five years of data after the SSOC policies were in effect, the brain imaging rate was 5 PP higher than the predicted outcome (Figure 3C). The 90% CI for the average discrepancy five years post policy does not cross zero, providing considerable support for the estimated effect. Most of the 90% CIs for the average at three and four years support the same conclusion. However, the evidence of adequate model fit before policies were in effect is limited, as discrepancies larger than 2.5% in any direction are likely.

### 4.3. In-Hospital Stroke Costs

Twenty-two states had between three and nine years of hospital cost data starting one year after SSOC policies were in effect. In the first quartile of 22 states with at least three years of data, the average observed costs per hospital stay among stroke patients was USD 610 less per year than predicted in the absence of any SSOC policies (Figure 3A). Costs for the second quartile of 12 states with at least seven years of cost data were USD 1252 less than predicted (Figure 3B). Among seven states in the third quartile with at least nine years’ data, costs were USD 1724 less per hospital stay than predicted (Figure 3C). In each quartile, the 90% CI, while somewhat wide, does not include zero, indicating strong evidence of the estimated effect. For the first quartile, the evidence of adequate model fit before SSOC policies took effect was relatively strong.

### 4.4. Age-Adjusted Stroke Mortality

Thirty-two states had between five and 11 years of stroke mortality data starting one year after SSOC policies were in effect. The first quartile of 32 states with at least five years of mortality data had on average one fewer stroke deaths per 100,000 population than predicted in the absence of SSOC policy interventions (Figure 4A). The second quartile of 21 states had at least eight years of mortality data and 1.4 fewer stroke deaths per 100,000 population than predicted without SSOC policies (Figure 4B). Ten states comprising the third quartile with at least eleven years of mortality data had 1.6 fewer deaths per 100,000 than predicted (Figure 4C). Point estimates and most of the 90% CIs support these effects, suggesting a null effect was unlikely, particularly among the ten states that were the earliest adopters. Further, the probability of a residual or discrepancy larger than 2.5% in any direction was extremely low, providing strong evidence of the model fit before SSOC policies were in effect.

### 4.5. In-Hospital Stroke Mortality

Nineteen states had between four and eleven years of in-hospital stroke mortality data starting one year after SSOC policies were in effect. Nineteen states in the first quartile with at least four years of outcome data had on average 2 more deaths per 1000 stroke inpatients than predicted in the absence of SSOC policies (Appendix A). Sixteen states in the second quartile with at least six years of outcome data after policies were in effect had on average 1 more death per 1000 stroke patients than predicted (Appendix A). However, in the third quartile of seven states with at least eleven years of outcome data following the SSOC policies, inpatient stroke mortality was lower than predicted, with 5 fewer deaths per 1000 inpatients over the period. Despite the smaller number of states with at least eleven years of outcome data, the effect in these states is stronger, as the 90% CI does not cross zero for the states with more years of data after policies were in effect. The evidence of model fit before any policy change was limited (i.e., the probability of a residual or discrepancy larger than 2.5% in any direction was between 10% and 20%) which may be due to the volatility in this outcome in some states. Results for two additional metrics of health disparities in stroke mortality are reported in the Appendix A.

## 5. Discussion

The results from this study suggest that states with one or more state-level SSOC policies in effect achieved higher proportions of certified PSC hospitals than they would have achieved without SSOC policies across three segments of states examined (each reflecting a different number of years of outcome data after policies were in effect). States with SSOC policies exhibited lower average in-hospital stroke costs and age-adjusted stroke mortality, although findings across different segments of states were mixed. Additionally, states with one or more state-level SSOC policies had higher percentages of stroke patients that received brain imaging results within 45 min of emergency department arrival and lower in-hospital stroke mortality than they would have achieved without SSOC policies. However, for these latter outcomes, evidence of model fit before policy change was limited. In contrast, SSOC policies did not appear to impact the absolute difference in stroke mortality between White and non-White populations or between rural and non-rural counties. These findings address outcomes that pertain to quality and efficiency of stroke care, as well as health outcomes in alignment with the intent of recommendations and funding provided by ASA, BAC, and CDC’s Coverdell Program [3,8,9,10]. In particular, the recommended integration of pre-hospital and in-hospital care provides a framework demonstrating the potential for enhanced access to quality stroke care at lowered costs, a relevant finding given that high costs of healthcare have led to hospital closures, primarily in rural areas [33].

These findings complement recent recommendations noting that establishing SSOC policies, such as tiered stroke center certification, triage and transport of stroke patients to PSCs or appropriate stroke facilities, pre-hospital notification, and inter-facility transfer of stroke patients to PSCs result in improved health and healthcare outcomes for patients [3]. These recommendations help clarify how SSOC laws relate to improvements in the efficiency and effectiveness of clinical care and subsequent outcomes. They also highlight the complexity of the relationships between program and policy. For example, tiered stroke certification and requirements that EMS provide triage and transport stroke patients to PSCs may incentivize hospitals to become certified, leading to increased numbers of PSCs and subsequent improvements in care and outcomes. Additionally, policy changes have occurred alongside demonstrated improvements in care such as improved timeliness and administration of thrombolytic therapy associated with the implementation of AHA/ASA GWTG, and the increased linkage of stroke patient data across care settings and improved stroke care practices demonstrated by CDC’s Coverdell Program [11,12].

Given that experimental methods are not feasible for examining state law impacts, we used a natural experimental design paired with longitudinal modeling to estimate having one or more SSOC policies in effect on outcomes. Results are not generalizable. The counterfactual analysis was designed to ensure that policy interventions were in effect for at least a full year before accounting for policy implementation. We also controlled for measured and non-measured covariates to ensure that differences observed between the actual outcomes and predicted outcomes are due to state SSOC policy interventions and not to other known differences between states. Relatively consistent results across outcomes from different data sources with different sets of strengths and limitations add credibility to the findings. In addition, consistency across groups of states when looking across three different lengths of time after SSOC policies were in effect raises further confidence in the findings.

## 6. Implications and Future Directions

This is the first study to establish that having at least one SSOC policy in effect leads to improvements in stroke outcomes. These findings can be used by public health practitioners and related audiences to build support for efforts to improve SSOCs within their jurisdictions. Additionally, they can be used by regional and local stakeholders including EMS agencies and dispatchers, health care providers, patients, caregivers, hospitals, home health companies, regulatory agencies, and payers to build, improve, and implement SSOC interventions.

While this study clarifies the benefits of state SSOC laws, there are remaining gaps in the knowledgebase. This study provides evidence that having one or more SSOC policy interventions in effect positively impacts quality and efficiency of stroke care and stroke health outcomes; further study is needed to identify and compare the relative contribution of specific SSOC policy interventions on these outcomes and other measures of patient function/disability. In addition, mixed methods research examining successful implementation and enforcement of SSOC policies will improve our understanding of how these mechanisms relate to outcomes. Finally, additional study to examine variations in stroke laws that authorize versus mandate SSOC policies could provide a more nuanced understanding of SSOC policies.

## 7. Study Limitations

Other health-related laws such as laws providing better funding for hospitals or incentivizing better post-hospital care, as well as stroke policies passed at the county, city, and township level (not considered in this study), may contribute to stroke-related outcomes. If such policies went into effect after SSOC policy changes and differentially affect the states where SSOC laws were in effect (for example, if these policies were implemented in tandem), then our estimates may overestimate the true effect of the SSOC laws. Conversely, if other health-related policies are more common in states without SSOC laws in effect, we could be underestimating the effect of SSOC laws. Additionally, differences in state-level SSOC policy implementation and enforcement could lead to variability in outcomes unaccounted for in our model.

The outcomes examined in this study were limited to those with accessible data aggregated at the state level with a sufficient history for the analysis. As such, we do not address relevant patient function/disability outcomes. Some outcomes that were accessible still had some degree of missing data as well as limited historical data available to inform the counterfactual analysis. Further, the in-hospital cost metric was not adjusted for inflation and, therefore, warrants additional caution when interpreting findings.

Our analysis accounted for a breadth of state-specific trends and programmatic con- textual factors that may influence observed outcomes to mitigate potential confounders. However, we cannot be certain that we adjusted for all possible factors. Further, our ability to control for measured and unmeasured confounders was limited by the number of years of available historical information for each outcome.

## 8. Conclusions

Annually more than 795,000 people in the US experience a stroke [2], requiring immediate, high-quality emergency care to prevent death and disability. Many states have applied policy interventions to improve access to lifesaving, time-sensitive treatment for stroke. Using a natural experimental design and modeling techniques, this study provides evidence that state policy interventions contributed to improved access to PSCs while outcomes including lower inpatient hospital costs and age-adjusted stroke mortality were mixed, and stroke patients that received brain imaging results within 45 min of emergency department arrival and in-hospital stroke mortality were improved but had limited evidence of model fit before policy change. Public health practitioners can use these findings to inform decision making when enhancing stroke systems of care to improve public health impacts.

## Figures and Tables

**Figure 1 healthcare-11-02842-f001:**
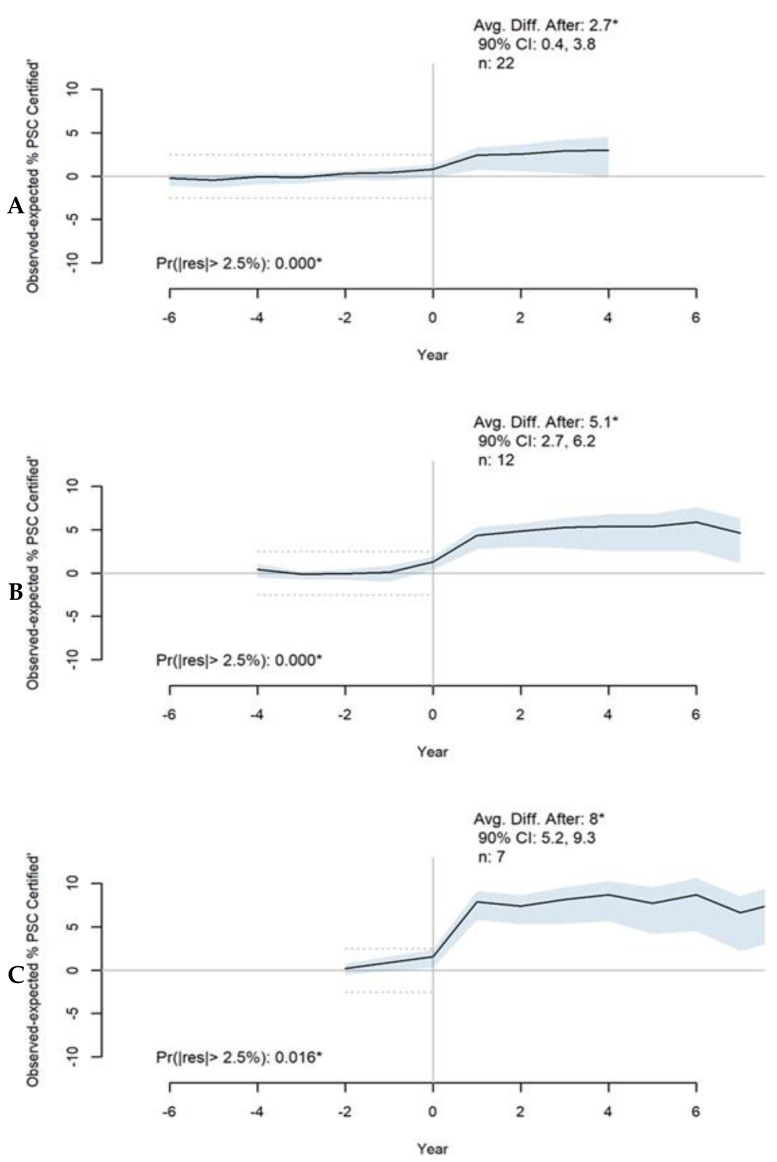
Average estimated effect of SSOC (stroke system of care) policies on PSC (primary stroke center) certification rates for three percentiles of states: (**A**) 25th percentile includes 22 states with 4 years of observed outcomes after policies were in effect; (**B**) 50th percentile includes 12 states with 7 years of observed outcomes after policies were in effect; (**C**) 75th percentile includes 7 states with 9 years of observed outcomes after policies were in effect. Year “0” on the x-axis is the year the first SSOC policy was in effect per state. The thick black line represents the average difference between observed and predicted outcome. The blue shaded area represents the 90% credible interval (CI) around the estimated effect. When the “Avg. Diff. After” value is marked with an asterisk, this indicates that the CI does not cross zero. “Pr(|res| > 2.5%” indicates the posterior probability of observing a discrepancy that is larger than +/−2.5% between the predicted and observed values (noted by the dashed lines); when marked with an asterisk, the probability is less than 0.05. These notations apply to all remaining figures.

**Figure 2 healthcare-11-02842-f002:**
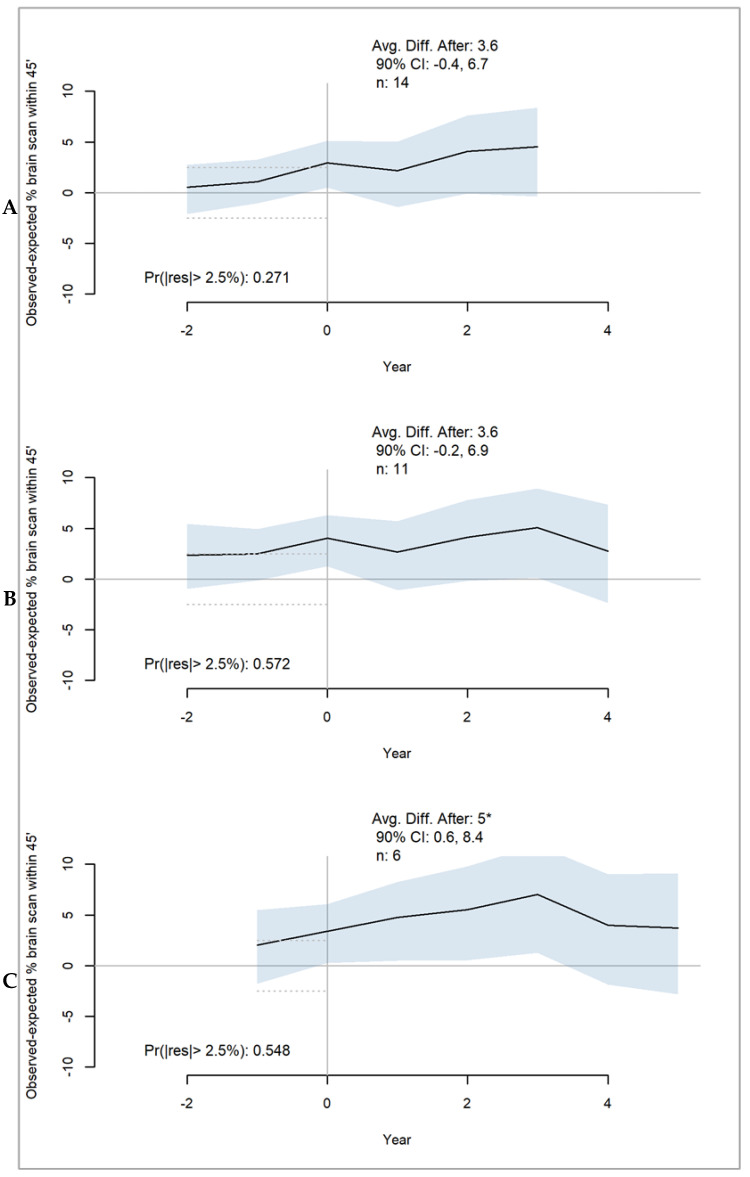
Average estimated effect of SSOC (stroke system of care) policies on brain imaging rates for three percentiles of states: (**A**) 25th percentile includes 14 states with 3 years of observed outcomes after policies were in effect; (**B**) 50th percentile includes 11 states with 4 years of observed outcomes after policies were in effect; (**C**) 75th percentile includes 6 states with 5 years of observed outcomes after policies were in effect. When the “Avg. Diff. After” value is marked with an asterisk, this indicates that the CI does not cross zero. “Pr(|res|> 2.5%” indicates the posterior probability of observing a discrepancy that is larger than +/−2.5% between the predicted and observed values (noted by the dashed lines); when marked with an asterisk, the probability is less than 0.05.

**Figure 3 healthcare-11-02842-f003:**
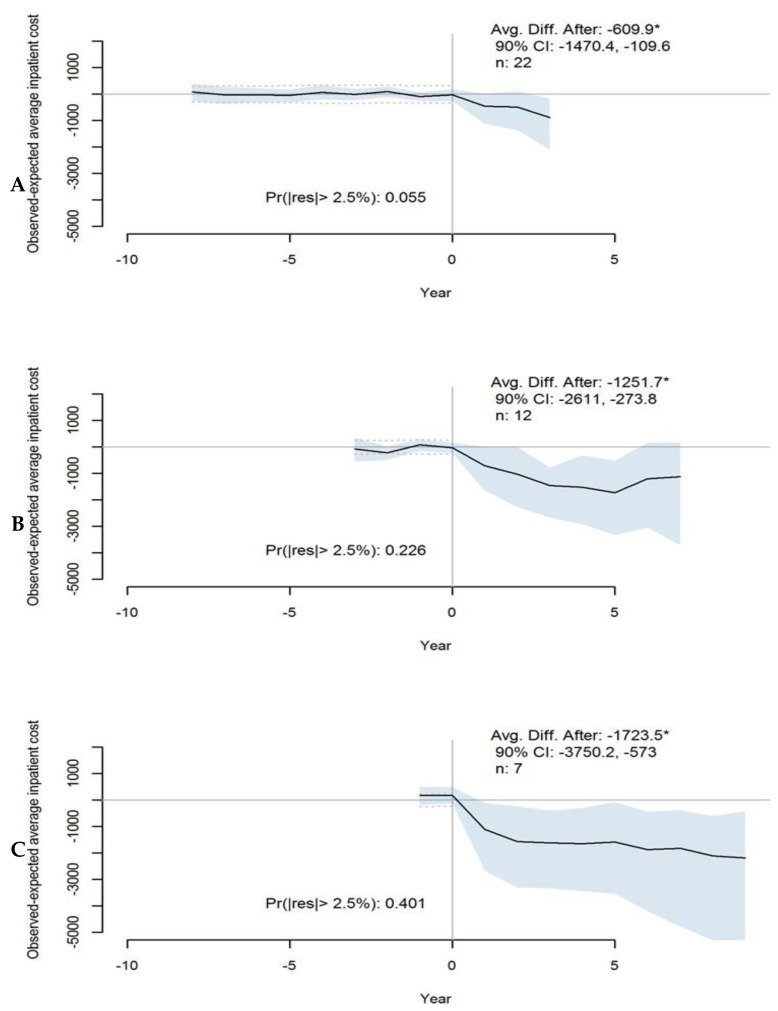
Average estimated effect of SSOC (stroke system of care) policies on in-hospital costs among stroke patients for three percentiles of states: (**A**) 25th percentile includes 22 states with 3 years of observed outcomes after policies were in effect; (**B**) 50th percentile includes 12 states with 7 years of observed outcomes after policies were in effect; (**C**) 75th percentile includes 7 states with 9 years of observed outcomes after policies were in effect. When the “Avg. Diff. After” value is marked with an asterisk, this indicates that the CI does not cross zero. “Pr(|res|> 2.5%” indicates the posterior probability of observing a discrepancy that is larger than +/−2.5% between the predicted and observed values (noted by the dashed lines); when marked with an asterisk, the probability is less than 0.05.

**Figure 4 healthcare-11-02842-f004:**
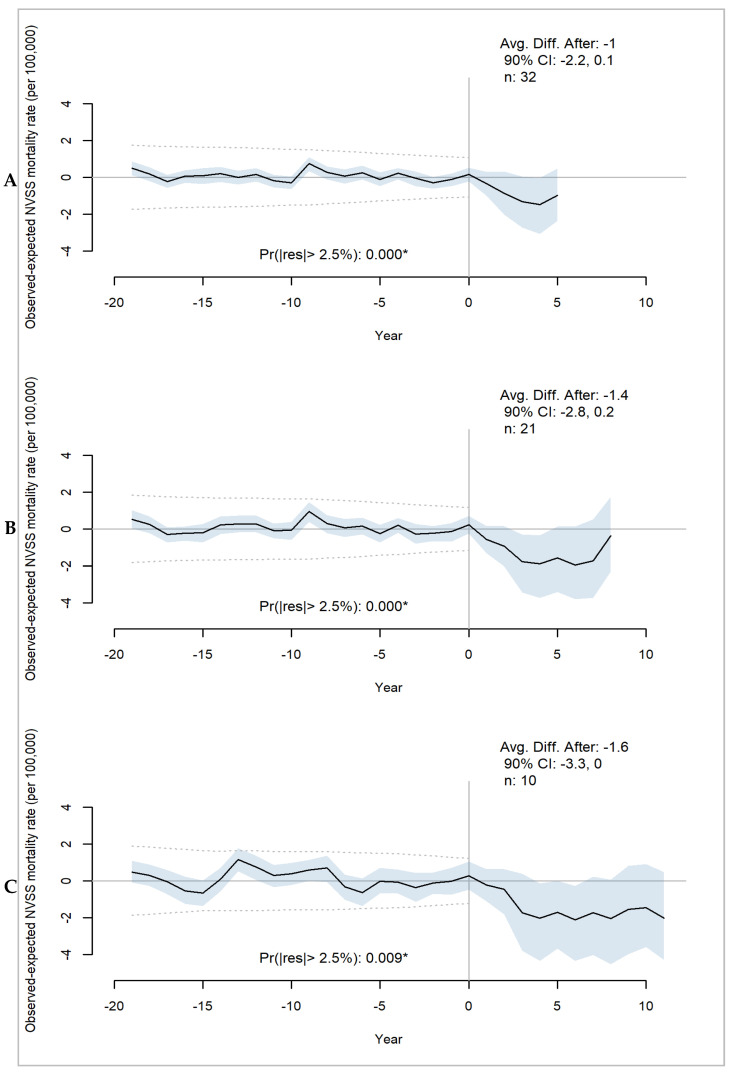
Average estimated effect of SSOC (stroke system of care) policies on age-adjusted stroke mortality for three percentiles of states: (**A**) 25th percentile includes 32 states with 5 years of observed outcomes after policies were in effect; (**B**) 50th percentile includes 21 states with 8 years of observed outcomes after policies were in effect; (**C**) 75th percentile includes 10 states with 11 years of observed outcomes after policies were in effect. When the “Avg. Diff. After” value is marked with an asterisk, this indicates that the CI does not cross zero. “Pr(|res|> 2.5%” indicates the posterior probability of observing a discrepancy that is larger than +/−2.5% between the predicted and observed values (noted by the dashed lines); when marked with an asterisk, the probability is less than 0.05.

**Table 1 healthcare-11-02842-t001:** SSOC ^1^ Policy interventions, descriptions, and number of states with policy in effect.

SSOC Policy Intervention	Number of States with Policy in Effect as of 2018
SSOC task force	20
Prehospital policy interventions	
EMS ^2^ stroke pre notification of receiving facility	6
EMS transport protocols to appropriate facility	18
EMS stroke assessment protocols	20
Standardized EMS stroke assessment	11
EMS air medical transport protocols	2
Inter-facility transfer agreements	17
EMS provider education in stroke assessment	10
Statewide or regional EMSS stroke care CQI ^3^ planning	8
In-hospital policy interventions	
Stroke center tiered approach	36
Authorizes acute stroke care through telemedicine	10
Statewide stroke CQI data system	16
Stroke centers report CQI stroke performance data	24
Recognizes nationally accredited PSC ^4^	30
Recognizes PSCs meeting state standards	5
Recognizes nationally accredited CSC ^5^	28
Recognizes CSCs meeting state standards	5
Recognizes nationally accredited ASRH ^6^	21
Recognizes ASRH meeting state standards	8

^1^ Stroke System of Care, ^2^ Emergency Medical Services, ^3^ Continuous Quality Improvement, ^4^ Primary Stroke Center, ^5^ Comprehensive Stroke Center, ^6^ Acute Stroke Ready Hospital.

## Data Availability

The data presented in this study are available on request from the corresponding author. The data are not publicly available due to use of some proprietary data.

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
