# Peer review of "Impact of State Stroke Systems of Care Laws on Stroke Outcomes"

_healthcare, 2023, doi:10.3390/healthcare11212842_

Round 1

Reviewer 1 Report

It has been my pleasure to review the manuscript entitled “Impact of State Stroke Systems of Care Laws on Stroke Out-comes”. The authors used an experimental design paired with longitudinal modeling to estimate the impact of having one or more SSOC policies in effect on outcomes.

My comments are as follows:

1. Abstract: It will be enhanced by providing actual data, not just descriptions.

2. Methods: One of the significant impacts of stroke legislation is to allow state to designate stroke centers. They are counted as stroke centers in real practice. Therefore, state certified stroke centers should be counted, or at least in a sensitivity analysis.

3.  Methods/Measured Covariates: Have these literatures been validated to be associated with outcomes? Patient comorbidities are validated factors associated with outcomes. Why were they not adjusted for?

4. How was the secular trend and other parallel quality initiatives accounted for, such as Target: Stroke?

5. stroke patients that received brain imaging results within 45 minutes of emergency department arrival: Has the conformity with this measure been associated with better outcomes?

6. Was there continuous improvement with the state legislation? Or changes occurred during the first year or two only?

7.      The author lumped all stroke legislations together. It is impossible for a state to have all of them. Which component of the stroke legislation has the actual impact on outcomes?

Reviewer 2 Report

This paper reports on an analysis comparing stroke outcomes in states that have stroke care laws vs those without. The authors found, in general positive effects.
My chief concern is authors attributing the outcomes to the laws (intervention). While I agree the laws are likely able to provide an environment that allows for better stroke care services and thus possibly better outcomes, in the absence of a randomised controlled trial that allows for correction for possible confounders, the conclusions, though seemingly encouraging, are at best hypothesis generating. The lack of a consistent effect on all outcome measures in all states is a case in point. Just as there are factors that the authors could point to to explain this, not all differences are adequately explicable. Different policies may have differing impacts, if any, and differing  combinations like-wise, at the state and possibly at the sub-state level. The text should reflect this cause-effect uncertainty.
Page 5 - ‘After considering all relevant historical information on each outcome as well as measured and unmeasured covariates, we found no systematic differences between states other than the SSOC policy interventions that are in effect. To that extent, the difference between the observed and the predicted outcomes measure the effect of having any SSOC policy intervention in place’ – the data that allows for this conclusion needs to be provided; I fear this conclusion may be an over-simplification of cause-effect. Still, i note para 3 of the Discussion, and the authors’ assessment of the study limitations.
I note the lack of sustained effect as seen by the trends in Fig2 B, 2C, 3A 3B, and the CIs consistent with no effect
Still, the authors excellent work should be recognized.
Minor issue - table 1 – I suggest that the headings be aligned such that SOCC task force, Prehospital policy interventions, and In-hospital policy interventions be aligned left, while the subsequent be spaced with a tab
